# Centralised RECIST Assessment and Clinical Outcomes with Lenvatinib Monotherapy in Recurrent and Metastatic Adenoid Cystic Carcinoma

**DOI:** 10.3390/cancers13174336

**Published:** 2021-08-27

**Authors:** Laura Feeney, Yatin Jain, Matthew Beasley, Oliver Donnelly, Anthony Kong, Rafael Moleron, Chandran Nallathambi, Martin Rolles, Paul Sanghera, Aung Tin, Danny Ulahannan, Harriet S. Walter, Richard Webster, Robert Metcalf

**Affiliations:** 1Department of Medical Oncology, The Christie NHS Foundation Trust, Manchester M20 4BX, UK; laura.feeney@nhs.net (L.F.); yatin.jain@nhs.net (Y.J.); 2Department of Oncology, University Hospitals Bristol NHS Foundation Trust, Bristol BS1 3NU, UK; matthew.beasley@uhbristol.nhs.uk; 3Department of Oncology, Portsmouth Hospitals University NHS Trust, Portsmouth PO6 3LY, UK; o.donnelly@nhs.net; 4Department of Oncology, Guys’ Campus, King’s College London, London SE5 9RS, UK; anthony.kong@kcl.ac.uk; 5Department of Oncology, University Hospital Birmingham NHS Foundation Trust, Birmingham B15 2TH, UK; Paul.Sanghera@uhb.nhs.uk; 6Department of Oncology, Aberdeen Royal Infirmary NHS Grampian, Aberdeen AB25 5ZN, UK; rafael.moleron@nhs.scot; 7Department of Oncology, Leeds Teaching Hospitals NHS Trust, Leeds LS9 7TF, UK; chandran.nallathambi@nhs.net (C.N.); dannyulahannan@nhs.net (D.U.); 8Department of Oncology, Swansea Bay University Health Board, Port Talbot SA12 7BR, UK; martin.rolles@wale.nhs.uk; 9Department of Oncology, The James Cook Cancer Institute, The James Cook University Hospital, Middlesbrough TS4 3BW, UK; aung.tin@nhs.net; 10Department of Oncology, University Hospitals of Leicester NHS Trust, Leicester LE1 5WW, UK; hw191@leicester.ac.uk; 11Department of Oncology, Velindre University NHS Trust, Cardiff CF15 7QZ, UK; Richard.Webster@wales.nhs.uk

**Keywords:** adenoid cystic carcinoma, salivary gland cancer, lenvatinib

## Abstract

**Simple Summary:**

Adenoid cystic carcinoma (ACC) is a rare cancer of the head and neck. Initial treatment may involve surgery and/or radiotherapy with the aim of removing the cancer and preventing spread to other parts of the body. In patients in whom ACC has recurred or spread, systemic therapies such as chemotherapy or immunotherapy have been shown to have minimal benefit and there are currently no recommended standard systemic treatment options. More recently, the targeted therapy lenvatinib has shown promising results in treating ACC patients. We aimed to summarise the real-world experience of lenvatinib use in ACC patients in the UK and found that although some patients obtained clinical benefit, there were no significant responses on radiological imaging by centralized assessment.

**Abstract:**

Adenoid cystic carcinoma (ACC) is a rare cancer of secretory glands. Recurrent or metastatic (R/M) ACC is generally considered resistant to cytotoxic chemotherapy. Recent phase II studies have reported improved objective response rates (ORR) with the use of the multi-kinase inhibitor lenvatinib. We sought to evaluate real-world experience of R/M ACC patients treated with lenvatinib monotherapy within the UK National Health Service (NHS) to determine the response rates by Response Evaluation Criteria of Solid Tumour (RECIST) and clinical outcomes. Twenty-three R/M ACC patients from eleven cancer centres were included. All treatment assessments for clinical decision making related to drug therapy were undertaken at the local oncology centre. Central radiology review was performed by an independent clinical trial radiologist and blinded to the clinical decision making. In contrast to previously reported ORR of 12–15%, complete or partial response was not observed in any patients. Eleven patients (52.4%) had stable disease and 5 patients (23.8%) had progression of disease as the best overall response. The median time on treatment was 4 months and the median survival from discontinuation was 1 month. The median PFS and OS from treatment initiation were 4.5 months and 12 months respectively. Multicentre collaborative studies such as this are required to evaluate rare cancers with no recommended standard of care therapy and variable disease courses.

## 1. Introduction

Adenoid cystic carcinoma (ACC) is a rare cancer of secretory glands that most often originates in the salivary glands and accounts for approximately 1% of all head and neck cancers [1] Rarely, ACC can arise from other sites, such as the trachea, bronchus, oesophagus, lacrimal gland, skin and breast. Surgical resection with or without adjuvant radiotherapy is considered for localised disease. However loco-regional and distant recurrence is common and may occur many years later [2]. Recurrent or metastatic (R/M) ACC is generally incurable and systemic therapies have been found to have low response rates [3,4].

The myeloblastosis (MYB) transcription factor regulates multiple transcriptional pathways including cellular differentiation and proliferation. Alterations in the MYB signalling pathway are considered to be a hallmark of ACC [5]. Previous studies have suggested that MYB overexpression may lead to up-regulation of several growth and angiogenic factors contributing to the autocrine activation of the fibroblast growth factor receptor (FGFR) and vascular endothelial growth factor receptor (VEGFR)-mediated angiogenesis [6,7,8]. Whole-genome sequencing of patient ACC samples has identified mutations in genes involved in the FGF/IGF/PI3K pathway [9]. However multi-kinase inhibitors (MKI) targeting this pathway, such as sorafenib [10,11,12,13] and axitinib [14,15], have shown minimal efficacy in treating patients with ACC. Other targeted therapies, such as imatinib [16], gefitinib [17], dasatinib [18], and dovitinib [19] have shown no significant activity assessed by RECIST (response valuation criteria in solid tumours) criteria.

Lenvatinib is a second-generation MKI with strong anti-angiogenic activity inhibiting VEGFR. In addition, it inhibits the receptor tyrosine kinases (RTK) FGF, PDGFRα, KIT and RET. These RTKs play a crucial role in tumour growth and progression.

Two single-arm phase II studies of lenvatinib reported RECIST response rates of 12% [20] and 15% [21] in a biomarker unselected population of R/M ACC patients. Although this is a higher signal than for MKIs in earlier studies of ACC patients, there is significant treatment-related toxicity with lenvatinib. The use of lenvatinib in R/M ACC has yet to be established as a standard of care.

We sought to evaluate the real-world experience of R/M ACC patients treated with lenvatinib through a named patient supply programme within the UK National Health Service (NHS) to determine the response rates by RECIST and clinical outcomes following initiation of lenvatinib monotherapy.

## 2. Materials and Methods

### 2.1. Patient Consent

Twenty-three R/M ACC patients from eleven cancer centres in the UK NHS provided informed consent to the collection of demographic, clinical, and genomic data. The study was granted research ethics approval under the MCRC Biobank Research Tissue Bank Ethics (NHS NW Research Ethics Committee 18/NW/0092) and was performed in accordance with the Declaration of Helsinki.

### 2.2. Patient Treatment

Patients received lenvatinib monotherapy through a named patient supply access programme with local NHS approvals. Treatment was administered and overseen at their local cancer centre. The target start dose was 24 mg once daily with dose reduction at clinician discretion. Subsequent dose reduction due to toxicity was at the discretion of the local treating team. Response assessments were conducted locally to inform decisions on drug continuation/discontinuation.

### 2.3. Clinical Assessments

All on treatment assessments for clinical decision making related to drug therapy were undertaken at the local oncology centre. In addition, all patients underwent centralised clinical review at The Christie NHS Foundation Trust (RM). Eligibility was confirmed prior to commencing lenvatinib and clinical follow up was undertaken throughout and after drug treatment. 

### 2.4. Radiology Assessments

All imaging was performed at the treating centre and clinical decisions were made on the basis of local radiological assessment and reporting. Images were then electronically transferred via the picture archiving and communication system (PACS) to The Christie NHS Foundation Trust. Central radiology review was performed by an independent clinical trial radiologist (YJ) and blinded to the clinical decision making using RECIST 1.1 [22]. Marker and non-marker lesions were identified and measured on the baseline imaging and measurements provided on all subsequent imaging to determine the percentage change in the sum of the marker lesions from baseline. The best overall response (BOR) was assessed as complete response (CR), partial response (PR), stable disease (SD), or progressive disease (PD). The objective response rate (ORR) corresponded to the sum of the CR and PR rates.

To further investigate for a signal of disease stabilisation following lenvatinib initiation, we calculated the RECIST measurements in the interval prior to initiation of therapy (pre-baseline) to contrast with any subsequent change whilst on therapy.

### 2.5. Genomic Profiling

For all patients, DNA was extracted from archival FFPE blocks and underwent next-generation sequencing using the Manchester Centre for Genomic Medicine National Health Service Genomics Laboratory Hub (Qiagen GeneRead DNAseq Targeted Panel V2) panel and/or commercially sourced next-generation sequencing (Roche, Foundation Medicine, Cambridge, MA, USA). Genetic alterations previously reported as having an association with prognosis, such as *TP53* loss of function, *NOTCH1* gain of function and *TERT* promoter mutation, were analysed against survival outcomes within this cohort.

### 2.6. Survival Analysis

Overall survival (OS) and progression-free survival (PFS) rates from treatment initiation were estimated using the Kaplan–Meier method and differences between groups were assessed using the log-rank test. Data were censored on 24 June 2021. OS was calculated from the date of lenvatinib initiation to the date of death from any cause. Patients who were alive at the last review were censored for OS analysis. PFS was calculated from the date of lenvatinib initiation to the date of clinical or radiological disease progression or death of any cause. Time on treatment was calculated from the date of lenvatinib initiation to the date of discontinuation as recorded in the medical notes.

Statistical analyses were conducted using the GraphPad Prism (Version 9.0, GraphPad Software, San Diego, CA, USA) and SPSS (version 27.0; IBM Corporation, Somer, NY, USA).

## 3. Results

### 3.1. Patient Characteristics and Prior Therapies

Baseline characteristics are summarised in Table 1. The median age at diagnosis was 46 years (range, 17 to 67 years), with 11 (47.8%) male and 12 (52.1%) female patients. Fifteen patients (65.2%) had ACC from a major salivary gland, with 8 (34.8%) from a minor salivary gland site. Twenty-one patients (91.3%) had lung metastases, 6 (26.0%) had bone metastases and 3 (13.0%) had nodal disease. Other sites of metastatic disease included skin (*n* = 1, 4.3%), brain (*n* = 1, 4.3%) and kidney (*n* = 1, 4.3%).

Prior to lenvatinib therapy, 17 patients (73.9%) received surgery and adjuvant radiotherapy, with 2 patients (8.7%) receiving surgery alone, one patient (4.3%) received chemoradiotherapy, one patient (4.3%) received palliative radiotherapy, and one patient (4.3%) was enrolled in a clinical trial as their primary treatment. Nine patients (39.1%) received palliative chemotherapy in the R/M setting prior to initiation of lenvatinib, of which 4 (44.4%) received two or more lines of systemic treatment. Four patients (17.3%) received prior systemic therapy within early phase clinical trials.

### 3.2. Treatment with Lenvatinib Monotherapy

All patients were treated with lenvatinib monotherapy. Nine patients (39.1%) commenced treatment at a starting dose of 24 mg once daily (Table 2), the remainder starting with a dose reduction due to either reduced performance status, impairment of hepatic function, or concern of potential toxicity. Of these nine, four required a dose reduction due to toxicity. Four patients (17.4%) discontinued treatment due to drug toxicity and 15 patients (65.2%) discontinued due to disease progression. The median duration of lenvatinib treatment was 4 months (range 1 to 22 months). Four patients (17.4%) remained on treatment at the time of analysis with a duration of treatment between 6 and 22 months.

### 3.3. Centralised Efficacy Assessment

The primary aim of this study was to determine radiological response rates, by RECIST criteria, with multikinase inhibition using lenvatinib monotherapy in R/M ACC to compare the real-world outcomes with the prior results of clinical trials in this setting. Two patients were excluded from centralised radiological review and efficacy assessment as all the original radiological images were not available for analysis. Contrasting with the 12–15% ORR by RECIST reported within clinical trials [20,21]. Complete or partial response was not observed in any patients. Eleven patients (52.4%) had stable disease and 5 patients (23.8%) had progression of disease as the best overall response (Table 3). Five patients (23.8%) discontinued treatment due to clinical progression or deterioration in performance status before the first radiological assessment on therapy was performed. 

To further assess for a signal of anti-tumour efficacy in this setting, individual responses were assessed (Figure 1A). Some degree of tumour regression of up to 12% of the sum of the target lesions was observed in 25% (4/16) of evaluable patients. We then assessed the percentage change in the sum of the marker lesions throughout the duration of therapy for the 16 patients who had at least one re-assessment scan on treatment (Figure 1B). Tumour shrinkage relative to the baseline was observed in four patients (19%). Of the two patients in whom minor reduction was seen exceeding 10%, one patient with a start dose of 24 mg had a reduction of 12% seen at reassessment imaging at one month, however, treatment was discontinued due to drug-related toxicity and decline in performance status and was not reinitiated. The second patient was treated with a starting dose of 14 mg and had the best reduction in marker lesions of 12% seen at the second reassessment imaging at six months. Treatment was continued without clinically significant toxicity and the patient remained on treatment for 20 months until discontinuation due to progressive disease. For all patients, the median time on treatment was 4 months (range 0 to 22 months) and the median survival from discontinuation was 1 month (range 0 to 15 months) (Figure 1C). At the time of the last follow up, four patients remained on treatment.

ACC typically shows indolent progression off therapy. To further investigate for a signal of disease stabilisation following lenvatinib initiation, we calculated the RECIST measurements in the interval prior to initiation of therapy (pre-baseline) to contrast with any subsequent change whilst on therapy (Figure 2). In patients showing an increase in tumour measurement from pre-baseline to baseline scans, the median percentage change was a 25% increase (range of 3 to 225%). In comparison, the median percentage change from baseline to first on treatment response scan for the same patients was a 3% increase (range of −12 to 35%). The median interval between pre-baseline and baseline imaging was 3 months (range 1 to 14 months) and between baseline and first reassessment was 3 months (range 1 to 9 months). In addition to the two cases of minor tumour regression described above (Figure 2(Aii,Cvii)), a further two cases in the 24 mg cohort (Figure 2(Aiv,Avi)) had evidence of tumour growth pre-lenvatinib followed by stabilisation of disease for 2 months and 1 month, respectively. These patients however discontinued treatment at this point due to clinical deterioration and subsequent death within weeks of lenvatinib discontinuation.

### 3.4. Progression Free Survival and Overall Survival with Lenvatinib Monotherapy

We next sought to determine PFS and OS from initiation of lenvatinib therapy in this real-world cohort. In contrast with previous studies which have reported a PFS of 9.1 and 17.5 months and OS of 27 months [20,21]. Figure 3A shows that the median PFS in this study was 4.5 months and 24% of patients were progression-free at 12 months. The median OS was 12 months (Figure 3B) and 51% were alive at 12 months.

DNA-based next-generation sequencing was available to classify patients as having *TP53* loss of function, *NOTCH1* gain of function or having *TERT* promoter mutation in 14/23 patients. A Kaplan–Meier estimate of OS was carried out (Figure 4). Consistent with previous reports in ACC [23], the presence of a *TP53* mutation was associated with a shorter median OS of 2 months compared to 13 months in those without a *TP53* mutation (*p* = 0.0007). Although statistical significance was not achieved, a shorter median OS appears to be associated with the presence of a *NOTCH1* mutation, 4.5 months versus 13 months (*p* = 0.3605). This is in keeping with previous studies demonstrating poor prognostic outcomes in ACC patients with *NOTCH* mutations [5,24]. Conversely, *TERT* mutations were associated with a longer median OS when compared to patients without a *TERT* mutation, 21 months versus 9 months (*p* = 0.0150), consistent with previous studies [5]. Further investigation in a larger cohort is required to determine whether these mutations are of value in predicting response to treatment with lenvatinib.

## 4. Discussion

In the present study of lenvatinib monotherapy in R/M ACC, objective radiological responses by RECIST were not observed in any patient. This is in contrast with the previously reported response rate of between 12 to 15% [20,21]. When considering any magnitude of tumour regression, one study reported tumour reduction in 66% (21/32) of patients, with 25% (8/32) showing a 20% or greater reduction in tumour size [21]. In this real-world analysis, any tumour reduction was observed in 4 patients (19%), with regression ranging from 1–12%. A possible explanation for the disparity in ORR observed in this study compared to the phase II studies, may be related to the starting dose of lenvatinib. In the previous phase II studies, patients commenced lenvatinib at the maximum dose level of 24 mg once daily. In the current study, 8 patients (38%) commenced lenvatinib at 24 mg once daily, of which 3 (37.5%) required a dose reduction during treatment. It is possible that this difference in dosing may have a negative impact on response rates. However, the majority of patients (71.9–85.7%) enrolled in the phase II studies required at least one dose reduction during treatment. In one study a dose reduction was required in 88% (21/24) of patients within the first 12 weeks of treatment [20]. Furthermore, only 4 patients maintained the full dose for a median of 1.7 months (range, 1.0–2.8 months).

In addition to the lenvatinib starting dose, other factors such as the timing of treatment during the disease course may also influence outcomes. As previously mentioned, R/M ACC can have an indolent course with long periods of stability leading to variable PFS. Given the relatively low response rate and significant toxicity associated with lenvatinib, a significant proportion of patients undergo surveillance with the initiation of systemic treatment in the event of clinically significant symptom or radiological progression. The median PFS (4.5 months) and OS (12 months) observed in this cohort were shorter than those reported in the previous phase II studies (median PFS range 9.1–17.5 months; median OS 27 months), which may be reflective of a later time in the disease course for this cohort compared to patients enrolled in clinical trials. This is consistent with the short survival from treatment discontinuation (median of one month) in the current study.

Given the rare nature and typically indolent disease course of R/M ACC, there is limited evidence to guide optimal systemic therapy management. Furthermore, there is no conclusive evidence that survival is prolonged by systemic therapy. This is an inherent issue for rare cancers such as ACC given the difficulty in enrolling patients and conducting clinical trials for conditions that affect relatively few individuals. Although there have been recent regulatory approvals for pan-TRK inhibition with entrectinib [25] or larotrectinib [26,27] in NTRK rearranged secretory salivary gland carcinoma, it can be difficult to provide the level of evidence required to obtain regulatory approval or incentivise the pharmaceutical industry to invest in this area. For example, in androgen receptor overexpressing salivary duct carcinoma, there are pre-clinical studies showing AR-dependency in cultured salivary duct carcinoma cell lines [28,29] and phase 2 data for combined androgen blockade [30] which is being adopted as a standard of care.

In recent years, alternative tyrosine kinase inhibitors (TKIs) have shown some promising results in the treatment of ACC patients. Apatinib, an oral TKI which is highly selective for and strongly inhibits vascular endothelial growth factor receptor 2 (VEGFR-2), has shown encouraging antitumour activity and tolerability in a phase II prospective study in the R/M ACC setting (NCT02775370) [31]. The study required radiological evidence of disease progression before enrollment to evaluate the contribution of apatinib to disease stabilisation. An ORR of 46.2% was reported which compares favourably with the ORR of 12–15% observed for lenvatinib as well as those reported in other TKIs trialled in R/M ACC, including axitinib [14,15], sorafenib [10,11,12,13], and sunitinib [32]. The median PFS was 19.7 months, and the median OS was not reached, both comparing favourably with lenvatinib and other TKIs [8,9,10,11,12,13,14,15,16,17,18,19]. Interestingly, patients who received a higher intensity of apatanib treatment had significantly longer PFS than those receiving a lower intensity (*p* = 0.024, HR 0.033, 95% CI 0.002–0.639). Serious adverse events were reported in 14.7% of patients and dose reduction was required in 76.9%. The outcome of a multi-centre prospective phase 2 study in R/M ACC to evaluate the efficacy and safety of rivoceranib, which has mainly overlapping tyrosine kinase inhibition with apatinib is awaiting (NCT04119453).

The findings of the current study are limited by the retrospective multi-centre design. As such, comprehensive information on unrelated cancer, clinical data, and adverse events experienced by patients during lenvatinib treatment were not available. Of the available data, heterogeneity in the baseline characteristics of patients was also observed. The purpose of this study was to provide a descriptive analysis of the real-world experience of lenvatinib use in R/M ACC patients. The limited number of patients included in the study was dictated by the low prevalence of ACC in the general population and thus a power analysis to determine the statistical significance of the results was not performed. Despite these limitations, multicentre collaborative studies such as this are required to evaluate rare cancers with no recommended standard of care therapy and variable disease courses.

## 5. Conclusions

This study showed that, in a real-world setting, lenvatinib monotherapy may offer a degree of disease stabilisation to patients with R/M ACC. However, this study has not found a strong signal of clinical benefit. In studies to date, multi-kinase inhibition remains the approach for treating R/M ACC, which has shown the greatest promise, and further evaluation of alternative agents with an improved efficacy and toxicity profile is warranted.

## Figures and Tables

**Figure 1 cancers-13-04336-f001:**
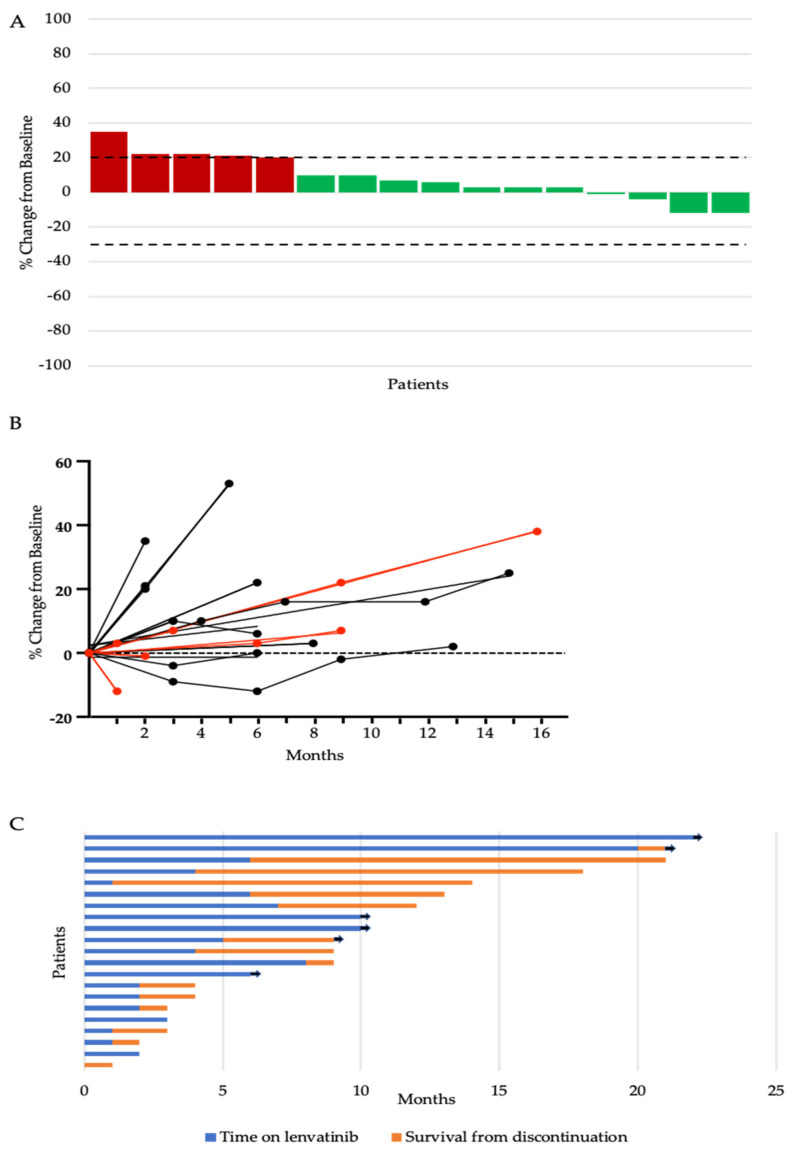
(**A**) Waterfall plot of maximum percent change in tumour size from baseline as measured by Response Evaluation Criteria in Solid Tumour (RECIST). The upper dotted line represents the threshold for progressive disease (a 20% increase in the sum of the longest diameter of the target lesions) and the lower dotted line represents the threshold for partial response (a 30% decrease in the sum of the longest diameter of the target lesions). (**B**) Change from the baseline (%) in the sum of the target lesions over time. Red lines represent patients on a starting dose of 24 mg. Black lines represent patients starting on a reduced dose. (**C**) Swimmers plot of time from the start of lenvatinib to the time of discontinuation (blue bar) and survival from discontinuation to death or last follow-up (orange bar). Each bar represents an individual patient, with the length of the bar corresponding to the time of overall survival. Arrow indicates patient is alive.

**Figure 2 cancers-13-04336-f002:**
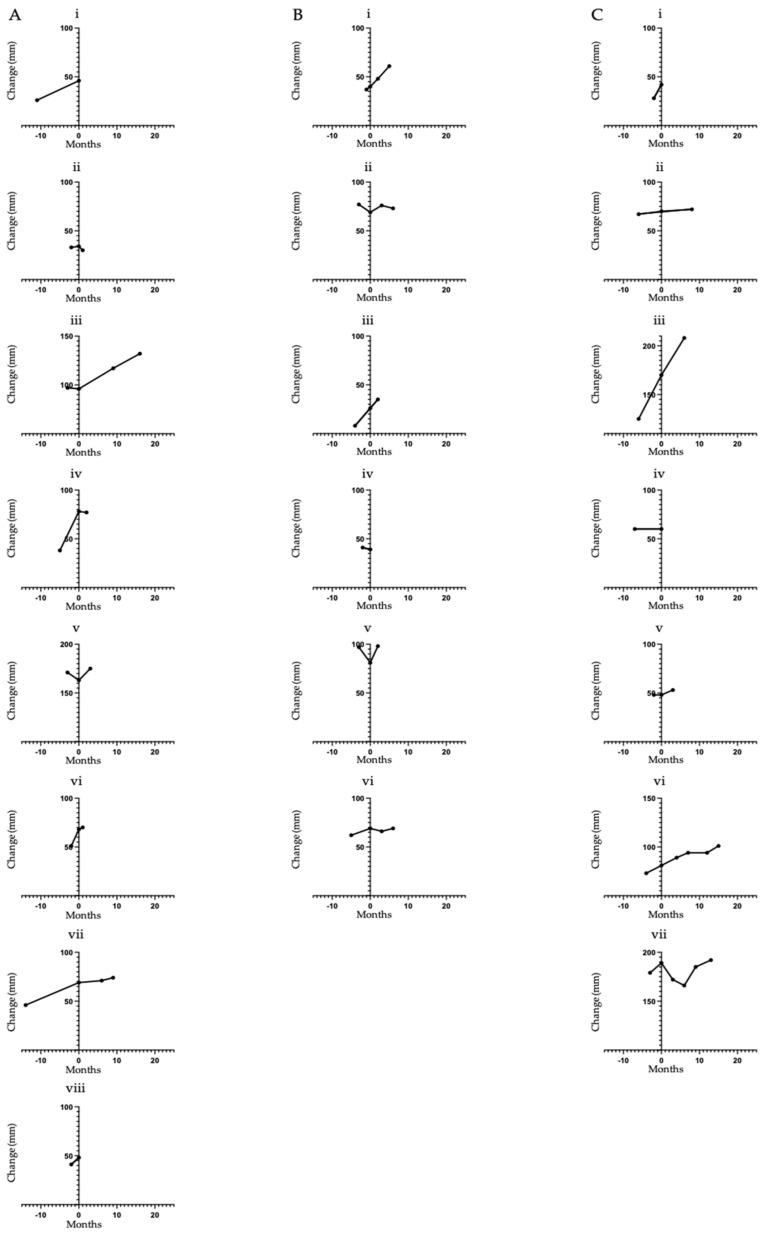
Change in the sum of target lesions over time (mm). (**A**) Patients initiated on lenvatinib 24 mg once daily. (**B**) Patients initiated on lenvatinib 20 mg once daily. (**C**) Patients initiated on lenvatinib 18 mg and 14 mg once daily. The Y axes are standardized using axis breaks to evaluate a 100 mm change in the measurements for comparison of the magnitude of change between patients. Individual patients are labelled i–viii.

**Figure 3 cancers-13-04336-f003:**
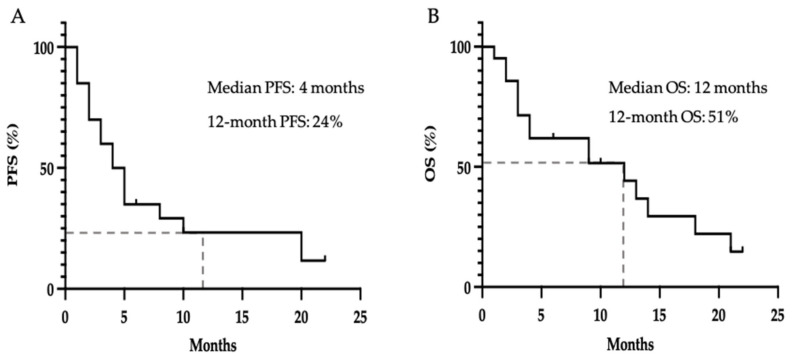
Kaplan–Meier estimate for (**A**) Progression free survival on lenvatinib, defined as time from first dose until objective disease progression; (**B**) Overall survival from initiation of lenvatinib, defined as time from first dose to death from any cause; dashes indicate censored events.

**Figure 4 cancers-13-04336-f004:**
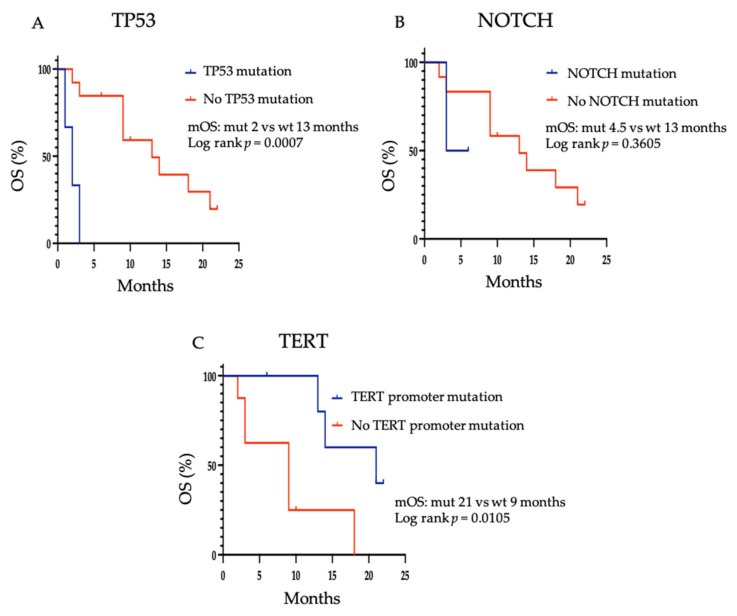
Kaplan–Meier estimate for overall survival in patients with or without (**A**) *TP53* mutation, (**B**) *NOTCH1* mutation and (**C**) *TERT* mutation. mOS = median overall survival; mut = mutant; wt = wild type.

**Table 1 cancers-13-04336-t001:** Baseline characteristics.

Characteristic	No. of Patients *n* = 23 (%) *
Age at diagnosis, years	
Median	46
Range	17 to 67
Sex	
Male	11 (47.8)
Female	12 (52.1)
Primary site of ACC	
Major salivary gland	15 (65.2)
Minor salivary gland	8 (34.8)
Metastatic disease site	
Lung	21 (91.3)
Bone	6 (26.0)
Skin	1 (4.3)
Brain	1 (4.3)
Kidney	1 (4.3)
Nodes	3 (13.0)
Primary therapy	
Surgery alone	2 (8.7)
Surgery + radiotherapy	17 (73.9)
Surgery + chemoradiotherapy	1 (4.3)
Chemoradiotherapy	1 (4.3)
Radiotherapy alone	1 (4.3)
Palliative systemic therapy	9 (39.1)
1 line	5/9 (55.6)
2+ lines	4/9 (44.4)
Clinical trial participation	4 (17.3)

* Values are numbers or percentages unless otherwise indicated.

**Table 2 cancers-13-04336-t002:** Lenvatinib starting dose and dose reduction.

Lenvatinib	No. of Patients (%)
Starting dose (once daily)	
24 mg	9 (39.1)
20 mg	7 (30.4)
18 mg	2 (8.9)
14 mg	5 (21.7)
Dose reduction on treatment (if started on)	
24 mg	4/9 (44.4)
20 mg	3/7 (42.9)
18 mg	0/2 (0)
14 mg	1/5 (20.0)
Reason for discontinuation	
Continues on treatment	4 (17.4)
Toxicity	4 (17.4)
Progressive disease	15 (65.2)

**Table 3 cancers-13-04336-t003:** RECIST response assessment to lenvatinib monotherapy.

Efficacy Measure (*n* = 21 Patients)	No. of Patients (%)
Best overall response	
Complete response	0
Partial response	0
Stable disease	11 (52.4)
Progressive disease	5 (23.8)
Off treatment before imaging	5 (23.8)

## Data Availability

The data presented in this study are available on request from the corresponding author. The data are not publicly available due to the requirement to uphold the data sharing with relevant approved researchers as stipulated in the ethical approval.

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
