# Peer review of "Centralised RECIST Assessment and Clinical Outcomes with Lenvatinib Monotherapy in Recurrent and Metastatic Adenoid Cystic Carcinoma"

_cancers, 2021, doi:10.3390/cancers13174336_

Round 1

Reviewer 1 Report

The manuscript of Feeney et al., on rare cancer,  evaluated the "real world" experience of R/M ACC patients treated with lenvatinib monotherapy within the UK National Health Service (NHS). The authors  determined the response rates by Response Evaluation Criteria of Solid Tumour (RECIST) and clinical outcomes for twenty-three R/M ACC patients from eleven cancer centers in UK. The methodology used was appropriate and despite the limitations of the study (mainly due to decreased number and different disease activity and prior treatment) reached different results than the previous reported in the literature. The authors should improve their manuscript supporting more cancer-unrelated clinical data (if any) from the patients medical records.   They should also make a comment on other studies on rare cancers treated with non recommended standard of care therapy. 

Author Response

Thank you to Reviewer 1 for their time taken to both assess this work and offer guidance on improvements. We believe their contributions have improved the quality of our manuscript, which we have now updated accordingly. Our specific responses to tReview 1's comments are detailed below.

Reviewer 1:

1. The authors should improve their manuscript supporting more cancer-unrelated clinical data (if any) from the patient’s medical records.  

We agree that this additional non-cancer related clinical information may enhance the reported findings. Unfortunately however, this data is not available as cancer-unrelated clinical data was not collected centrally through the Manchester Cancer Research Centre (MCRC) biobank. To reflect this consideration we have added an additional statement in the discussion regarding limitations (line 321 to 323).

“The findings of the current study are limited by the retrospective multi-centre design. As such, comprehensive information on cancer unrelated clinical data and adverse events experienced by patients during lenvatinib treatment were not available.”

2. They should also make a comment on other studies on rare cancers treated with non-recommended standard of care therapy. 

We thank Reviewer 1 for this suggestion and have added additional text to the discussion (please see line 295 to 305.)

“This is an inherent issue for rare cancers such as ACC given the difficulty in enrolling patients and conducting clinical trials for conditions that affect relatively few individuals. Although there have been  recent regulatory approvals for pan-TRK inhibition with entrectinib25 or larotrectinib26,27 in NTRK rearranged secretory salivary gland carcinoma, it can be difficult to provide the level of evidence required to obtain regulatory approval or incentivise pharmaceutical industry to invest in this area. For example in androgen receptor overexpressing salivary duct carcinoma there are pre-clinical studies showing AR-dependency in cultured salivary duct carcinoma cell lines28,29 and phase 2 data for combined androgen blockade30 which is being adopted as a standard of care.”  

Many thanks again for your consideration.

Reviewer 2 Report

This study evaluates the effectiveness of treating Adenoid Cystic Carcinoma (ACC) patients with the multikinase inhibitor lenvatinib. The study included 23 patients from 11 cancer centers in the UK. Unfortunately, the results are largely negative, as complete or partial response was not observed in any patients. Due to the many centers involved, and the small number of patients, there are several variables such as dose and timing of treatment that likely varied amongst the centers, decreasing the ability to make conclusions.

Although this study did not find any clinical benefit to the kinase inhibitor treatment, the authors still state the the drug may off a degree of disease stabilization. Thus, readers are left without the ability to draw any conclusions. The authors cannot say whether the drug helps or not.

Given their inability to make a conclusion, it would be prudent to re-evaluate the statistical analysis (i.e. power analysis) to see if this study really had sufficient statistical power.

Minor point: in the Introduction, the authors make some statements about ACC that are somewhat misleading, especially about the role of Myb activating the FGFR and VEGFR pathways, which has not been proven.

Author Response

Thank you to Reviewer 2 for their time taken to both assess this work and offer guidance on improvements. We believe their contributions have improved the quality of our manuscript, which we have now updated accordingly. Our specific responses to Reviewer 2's comments are detailed below.

Reviewer 2:

1. Given their inability to make a conclusion, it would be prudent to re-evaluate the statistical analysis (i.e. power analysis) to see if this study really had sufficient statistical power.

We are grateful to Reviewer 2 for this comment and have revisited the statistical plan. In a post-hoc power analysis calculation, 23 subjects were included in the study and the total number was restricted by the rare nature of ACC requiring national recruitment. The study would provide very limited power to detect a <15% difference in RECIST response rates between the study population and the published population in whom 15% response rates have been observed.  Eighty patients in the study group would be required to detect a difference of 5% between the current group and published population with >80% power and 0.05 alpha. To reflect this limitation, we have added the following additional statement added. Please see line 423 to 427.

“The purpose of this study was to provide a descriptive analysis of the real-world experience of lenvatinib use in R/M ACC patients. The limited number of patients included in the study was dictated by the low prevalence of ACC in the general population and thus a power analysis to determine statistical significance of the results was not performed.”

2. Minor point: in the Introduction, the authors make some statements about ACC that are somewhat misleading, especially about the role of MYB activating the FGFR and VEGFR pathways, which has not been proven.

Thank you for this correction, we have revised the wording to reflect the uncertainty around the link between MYB and FGFR/VEGFR (Please see line 67 to 71.)

“Previous studies have suggested that MYB overexpression may lead to up-regulation of several growth and angiogenic factors contributing to the autocrine activation of the fibroblast growth factor receptor (FGFR) and vascular endothelial growth factor receptor (VEGFR)-mediated angiogenesis.6,7,8

Many thanks again for your consideration.